# LEARNING TO TRANSLATE NOISE
# FOR ROBUST IMAGE DENOISING

## ABSTRACT

Image denoising techniques based on deep learning often struggle with poor generalization performance to out-of-distribution real-world noise. To tackle this challenge, we propose a novel noise translation framework that performs denoising on an image with translated noise rather than directly denoising an original noisy image. Specifically, our approach translates complex, unknown real-world noise into Gaussian noise, which is spatially uncorrelated and independent of image content, through a noise translation network. The translated noisy images are then processed by an image denoising network pretrained to effectively remove Gaussian noise, enabling robust and consistent denoising performance. We also design well-motivated loss functions and architectures for the noise translation network by leveraging the mathematical properties of Gaussian noise. Experimental results demonstrate that the proposed method substantially improves robustness and generalizability, outperforming state-of-the-art methods across diverse benchmarks.

## 1 INTRODUCTION

Image denoising aims to restore the pure signal from noisy images and serves as a critical preprocessing step to improve the visual quality of input images, extending the applicability of various downstream tasks. Recent advances in deep learning have significantly improved the performance of image denoising models (Zhang et al., 2017; 2018; Guo et al., 2019; Zamir et al., 2020; 2022b;a; Chen et al., 2022; Zhang et al., 2024). A common assumption in early approaches was that camera noise could be modeled as Gaussian noise (Mao et al., 2016; Zhang et al., 2017; 2018), which simplified the process of generating noisy-clean image pairs by adding synthetic Gaussian noise. This allowed for the creation of large datasets that could be used to train denoising models in a supervised manner, playing a crucial role in advancing the development of denoising models.

Although these models trained on synthetic dataset perform well under controlled environments, they often struggle to generalize to real-world scenarios due to the fundamental differences between synthetic and real noise distributions (Guo et al., 2019). In response, researchers have collected clean-noisy image pairs from real images (Abdelhamed et al., 2018; Xu et al., 2018; Yue et al., 2019) to address realistic noises, but models trained on this data still tend to overfit to the specific noise-signal correlations present in the training data. Capturing the full spectrum of noise distributions in real world images is impractical and even unrealistic.

To address this challenge, we propose a novel noise translation framework for image denoising to better generalize to diverse real-world noise using a limited training dataset. The intuition behind our framework is as follows. While existing denoising algorithms trained on images with Gaussian noise exhibit limited performance when applied to real noisy images, we observed that adding Gaussian noise to these noisy images significantly improves their effectiveness in denoising, as shown in Figure 1. This observation motivated us to explore the idea that, instead of directly denoising unseen real noise, first translating it into known Gaussian noise and then applying denoising could improve the model's ability to generalize across unseen and OOD noise. To this end, we introduce the lightweight *Noise Translation Network*, which, prior to the denoising process, utilizes Gaussian injection blocks to transform arbitrary complex noise into Gaussian noise that is spatially uncorrelated and independent of an input image. The translated images are then processed by the pretrained denoising networks specialized for Gaussian noise, resulting in the clean denoised images. Our ex-

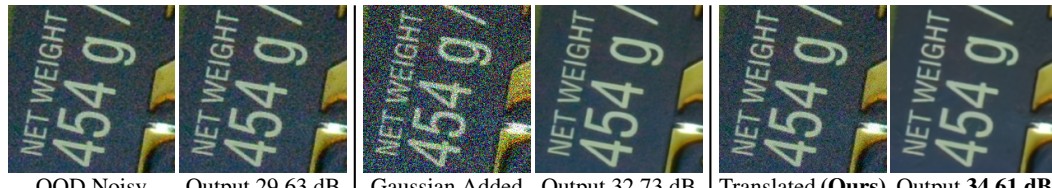

| OOD Noisy | Output 29.63 dB | Gaussian Added | Output 32.73 dB | Translated (**Ours**) | Output **34.61 dB** |

Figure 1: Observations on denoising network trained with synthetic Gaussian noise applied to a noisy image from the CC dataset. Left pair shows the original noisy input and its output, while the middle pair shows the input added with Gaussian noise and the corresponding output. Experiments were conducted using the Restormer with officially published model weights. Last pair shows the Gaussian-translated input and the resulting output of our method. The denoised outputs are evaluated with Peak Signal-to-Noise Ratio (PSNR↑) against the ground truth image. Zoom in for better details.

perimental results and analysis validate that the proposed framework outperforms existing denoising approaches by huge margins on various benchmarks.

Overall, our key contributions are summarized as follows:

- We propose a novel noise translation framework for robust image denoising, which converts unknown complex noise of the input image into Gaussian noise which is spatially uncorrelated and independent of the image content. The translated images are then processed by pretrained denoising networks specialized in removing Gaussian noise.
- We employ well-motivated loss functions and architecture for the noise translation network. The proposed approach guides the noise distribution of the input image to Gaussian distribution both implicitly and explicitly by rigorously leveraging the mathematical properties.
- We demonstrate the efficacy of our approach through extensive experiments on image denoising benchmarks with diverse noise distributions, achieving significant improvements in terms of robustness and generalization ability compared to existing methods.

The rest of this paper is organized as follows. Section 2 reviews the related literature. We present our noise translation framework for image denoising in Section 3 and demonstrate its effectiveness in Section 4. We conclude our paper in Section 5.

## 2 RELATED WORKS

**Image denoising** In recent years, deep learning has led to significant progress in image denoising, achieving impressive results by leveraging paired noisy and clean images for training. DnCNN (Zhang et al., 2017) pioneered the use of CNNs for image denoising, which paved the way for further advancements involving residual learning (Gu et al., 2019; Liu et al., 2019; Zhang et al., 2019), attention mechanisms (Liu et al., 2018; Zhang et al., 2019), and transformer models (Zamir et al., 2022a; Zhang et al., 2024). Despite its success, acquiring the noisy-clean pairs required for supervised training remains a significant challenge. To address this, self-supervised approaches (Lehtinen et al., 2018; Krull et al., 2019; Batson & Royer, 2019; Pang et al., 2021; Li et al., 2023) have emerged to train networks using only noisy images, but these models typically perform considerably worse than their supervised counterparts. Additionally, zero-shot approaches (Quan et al., 2020; Huang et al., 2021; Mansour & Heckel, 2023) have been proposed for image denoising even without training dataset, but they require substantial computational cost at inference, making them impractical for real-time applications. In contrast to these methods that aim to reduce dependency on supervised data, out approach leverages supervised data but focuses on achieving good generalization performance with a limited amount of data.

**Generalization for denoising** Generalization is a critical challenge in image denoising, as the performance of denoising models often degrades when encountering noise characteristics that were not seen during training. To handle unseen noise type or levels, DnCNN (Zhang et al., 2017) employed blind Gaussian training to adapt to various noise levels, while Mohan et al. (2020) designed a bias-free network to prevent overfitting to noise levels in the training set. More recent works employed masking-based learning (Chen et al., 2023) or leverage the pre-trained CLIP encoder (Cheng

et al., 2024) to prevent overfitting by encouraging the model to understand global context rather than relying on local patterns. While these approaches enhance robustness to unseen noise, they often struggle to produce high-quality image restoration, particularly in complex real-world scenarios.

To address real-world noise, researchers have focused on constructing training datasets that closely resemble real noise distribution. This includes collecting real clean-noise image pairs (Abdelhamed et al., 2018; Xu et al., 2018; Yue et al., 2019) and learning to generate realistic noise through data augmentation (Jang et al., 2021; Cai et al., 2021) or adversarial attacks (Yan et al., 2022; Ryou et al., 2024). However, these approaches are limited to the noise distributions represented in the training dataset and fail to generalize effectively to unseen OOD noise. Our approach overcomes this limitation by incorporating a noise translation process that transforms the complex distribution of real noise into a known Gaussian distribution, improving performance on OOD data significantly.

## 3 METHOD

In this section, we present a robust image denoising framework featuring a novel noise translation process, designed to effectively handle diverse unseen noise. Our framework consists of: (1) a noise translation network to transform the arbitrary noise in an input image into ideal Gaussian noise, and (2) a denoising network to remove the translated noise to produce a clean output.

### 3.1 IMAGE DENOISING NETWORK

First of all, we train a denoising network as follows. Our image denoising network aims to recover a clean image from a noisy input, which can be mathematically formulated as

$$\hat{I} = \mathcal{D}(I; \boldsymbol{\theta}), \tag{1}$$

where $\mathcal{D}(\cdot; \boldsymbol{\theta})$ denotes an image denoising network parameterized by $\boldsymbol{\theta}$, and $I, \hat{I} \in \mathbb{R}^{H \times W \times C}$ represents a noisy input image and its corresponding denoised output, respectively.

The goal of supervised training for our denoising network is to ensure that the denoised output $\hat{I}$, to closely match the ground-truth clean image $I_{\text{GT}}$. To achieve this, the model parameters $\boldsymbol{\theta}$ are optimized by minimizing the following loss function:

$$\mathcal{L} = \|\mathcal{D}(I; \boldsymbol{\theta}) - I_{\text{GT}}\|. \tag{2}$$

Our method is model agnostic, allowing us to use existing denoising models and focus only on how to effectively remove Gaussian noise. To train our image denoising network to eliminate Gaussian noise, we utilize a training dataset consisting of clean images paired with their corrupted versions with synthetic additive Gaussian noise. We additionally use Gaussian-augmented real noisy-clean image pairs, where the noisy images are further corrupted with Gaussian noise. In our framework, our image denoising network is optimized with the loss function in (2) to make it specialized in removing Gaussian noise.

### 3.2 NOISE TRANSLATION NETWORK

Figure 2 illustrates the overall training pipeline of the noise translation network. Formally, our framework first transforms a noisy image $I$ into an image with Gaussian noise $I_{\mathcal{T}}$, which is then fed into the denoising network to produce the final denoised output $\hat{I}_{\mathcal{T}}$, represented by

$$\hat{I}_{\mathcal{T}} = \mathcal{D}(I_{\mathcal{T}}; \boldsymbol{\theta}^*) = \mathcal{D}(\mathcal{T}(I; \boldsymbol{\phi}); \boldsymbol{\theta}^*), \tag{3}$$

where $\mathcal{T}(\cdot, \boldsymbol{\phi})$ denotes the noise translation network with parameters $\boldsymbol{\phi}$, and $\mathcal{D}(\cdot, \boldsymbol{\theta}^*)$ indicates the pretrained denoising network with parameters $\boldsymbol{\theta}^*$. Note that $\mathcal{D}(\cdot, \boldsymbol{\theta}^*)$ is specialized in handling Gaussian noise, and its parameters are fixed during training the noise translation network. We next discuss how to train the noise translation network $\mathcal{T}(\cdot, \boldsymbol{\phi})$ by providing the following two loss terms.

### 3.2.1 IMPLICIT NOISE TRANSLATION LOSS

Our goal is to transform an arbitrary noisy input image $I$ into a noise-translated image $I_{\mathcal{T}}$ that is well-suited for the pretrained denoising network, which is specialized for handling Gaussian noise.

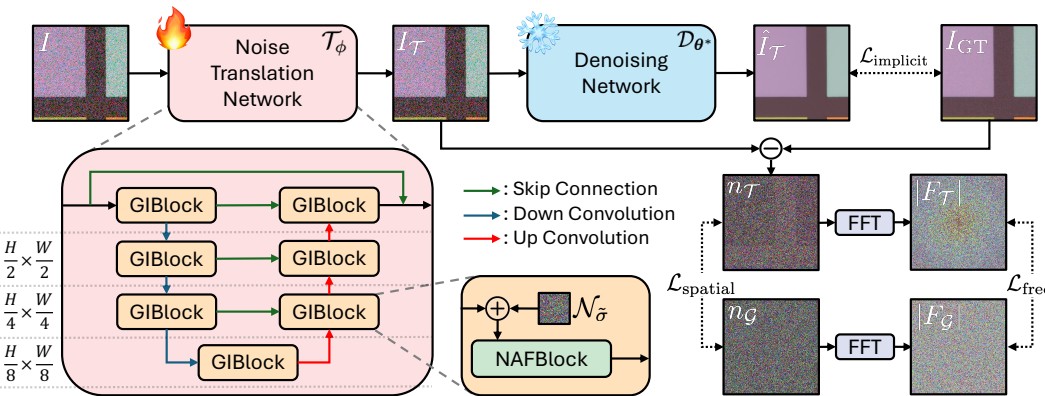

Figure 2: Illustration of our overall training framework, which includes the noise translation network and an existing denoising network specialized in handling Gaussian noise.

To achieve this, we optimize the noise translation network using a loss function, referred to as the *implicit* noise translation loss, which is designed to minimize the difference between the denoised image and the ground-truth clean image:

$$\mathcal{L}_{\text{implicit}} = \|\hat{I}_\mathcal{T} - I_{\text{GT}}\|_1 = \|\mathcal{D}(\mathcal{T}(I; \phi); \theta^*) - I_{\text{GT}}\|_1, \tag{4}$$

which guides the network to translate unseen noise into a form that the pretrained denoiser can handle effectively. A straightforward approach to train a noise translation network is to use real noisy-clean image pairs as $I$ and $I_{\text{GT}}$. To handle various noise levels in real-world scenarios that are lacking in the limited training set, we apply data augmentation by adding a random level of synthetic Gaussian noise to the noisy image $I$. This helps the noise translation network generalize more effectively across diverse noise conditions.

### 3.2.2 EXPLICIT NOISE TRANSLATION LOSS

The implicit noise translation loss helps to translate the image into a form preferable for the pre-trained denoising network, but it does not ensure that the noise distribution is transformed into an ideal Gaussian distribution, as it lacks direct control over the noise characteristics. To address this, we introduce an explicit loss function that directly guides the noise to follow Gaussian distribution.

Let $n_\mathcal{T} = I_\mathcal{T} - I_{GT} \in \mathbb{R}^{H \times W \times C}$ represent the translated noise, and let $n_\mathcal{G} \in \mathbb{R}^{H \times W \times C}$ be a random variable following a Gaussian distribution $\mathcal{N}(\hat{\mu}, \hat{\sigma})$, where $(\hat{\mu}, \hat{\sigma})$ denote the empirical mean and standard deviation calculated from all elements of $n_\mathcal{T}$. Our objective is to adjust the distribution of $n_\mathcal{T}$ to closely align with the distribution of $n_\mathcal{G}$. To achieve this, we utilize the Wasserstein distance to measure the difference between their distributions and employ it as a loss function to minimize:

$$\mathcal{L}_{\text{spatial}} \equiv d_{W_1}(n_\mathcal{T}, n_\mathcal{G}), \tag{5}$$

where $d_{W_1}(\cdot, \cdot)$ is 1-Wasserstein distance, also known as the Earth Mover's Distance. To calculate this, we first flatten each channel in $n_\mathcal{T}$ and $n_\mathcal{G}$ over the spatial dimensions into one-dimensional vectors, and then sort them in an ascending order. Let $\mathbf{X}^c \equiv (X_1^c, X_2^c, \ldots, X_{H \times W}^c)$ and $\mathbf{Y}^c \equiv (Y_1^c, Y_2^c, \ldots, Y_{H \times W}^c)$ denote the ordered values of $n_\mathcal{T}$ and $n_\mathcal{G}$ for the $c^{\text{th}}$ channel, respectively. The 1-Wasserstein distance is then calculated by the following simple function of the order statistics[1]:

$$d_{W_1}(n_\mathcal{T}, n_\mathcal{G}) = \frac{1}{H \cdot W \cdot C} \sum_{c=1}^{C} \sum_{i=1}^{H \cdot W} |X_{(i)}^c - Y_{(i)}^c|. \tag{6}$$

This loss encourages the translated noise $n_\mathcal{T}$ to follow a Gaussian distribution element-wise, but it is still insufficient to ensure that $n_\mathcal{T}$ is spatially uncorrelated. To handle the spatial correlation, we convert the signals of $n_\mathcal{T}$ and $n_\mathcal{G}$ into the frequency domain using their respective channel-wise

---

[1]Please refer to Section A for the detailed proof.

Fourier transforms, which are given by

$$F_{\mathcal{T}}^c(u, v) = \sum_{x=0}^{H-1} \sum_{y=0}^{W-1} n_{\mathcal{T}}(x, y, c) e^{-2\pi i \left( \frac{ux}{H} + \frac{vy}{W} \right)}, \tag{7}$$

$$F_{\mathcal{G}}^c(u, v) = \sum_{x=0}^{H-1} \sum_{y=0}^{W-1} n_{\mathcal{G}}(x, y, c) e^{-2\pi i \left( \frac{ux}{H} + \frac{vy}{W} \right)}, \tag{8}$$

where $(u, v)$ are the frequency domain coordinates and $c$ is a channel index. Since $n_{\mathcal{G}}$ is spatially uncorrelated Gaussian noise, the real and imaginary parts of the Fourier coefficients, $F_{\mathcal{G}}^c(u, v)$, also follow *i.i.d.* Gaussian distributions with zero mean and the same variance. Consequently, the magnitude of the Fourier coefficients, $|F_{\mathcal{G}}^c(u, v)|$, follows a Rayleigh distribution as

$$p_R(|F_{\mathcal{G}}^c(u, v)|; \sigma) = \frac{|F_{\mathcal{G}}^c(u, v)|}{\sigma^2} \exp\left( -\frac{|F_{\mathcal{G}}^c(u, v)|^2}{2\sigma^2} \right), \tag{9}$$

which implies that $|F_{\mathcal{T}}^c(u, v)|$ should also follow a Rayleigh distribution to ensure that $n_{\mathcal{T}}$ is spatially uncorrelated. To this end, similar to Eqs. (5) and (6), we minimize the difference between the distributions of $|F_{\mathcal{T}}^c(u, v)|$ and $|F_{\mathcal{G}}^c(u, v)|$ by utilizing 1-Wasserstein distance, which is defined as

$$\mathcal{L}_{\text{freq}} \equiv d_{W_1}(|F_{\mathcal{T}}|, |F_{\mathcal{G}}|) = \frac{1}{H \cdot W \cdot C} \sum_{c=1}^{C} \sum_{i=1}^{H \cdot W} |\tilde{X}_{(i)}^c - \tilde{Y}_{(i)}^c|, \tag{10}$$

where $\tilde{\mathbf{X}}^c \equiv (\tilde{X}_1^c, \tilde{X}_2^c, \ldots, \tilde{X}_{H \times W}^c)$ and $\tilde{\mathbf{Y}}^c \equiv (\tilde{Y}_1^c, \tilde{Y}_2^c, \ldots, \tilde{Y}_{H \times W}^c)$ are the sorted values of flattened magnitude of Fourier coefficients $|F_{\mathcal{T}}^c(u, v)|$ and $|F_{\mathcal{G}}^c(u, v)|$, respectively.

The full explicit noise translation loss is defined by $\mathcal{L}_{\text{spatial}}$ and $\mathcal{L}_{\text{freq}}$ as

$$\mathcal{L}_{\text{explicit}} = \mathcal{L}_{\text{spatial}} + \beta \cdot \mathcal{L}_{\text{freq}}, \tag{11}$$

where $\beta$ is a hyperparameter that balances the contribution of the two Wasserstein distances. This loss function explicitly guides the translated noise to follow Gaussian distribution.

The total loss function for training the noise translation network is given by

$$\mathcal{L}_{\text{total}} = \mathcal{L}_{\text{implicit}} + \alpha \cdot \mathcal{L}_{\text{explicit}}, \tag{12}$$

where $\alpha$ is a hyperparameter to control the influence of implicit and explicit loss terms.

### 3.2.3 GAUSSIAN INJECTION BLOCK

As illustrated in Figure 2, our noise translation network is built upon a lightweight U-Net architecture, where each layer is composed of Gaussian Injection Blocks (GIBlock). GIBlock incorporates a Nonlinear Activation-Free (NAF) block from NAFNet (Chen et al., 2022) along with our key idea to align the discrepancy between training and inference stage: Gaussian noise injection.

In the training stage of noise translation network, random levels of Gaussian noise is augmented to an input image $I$ to address diverse noise conditions, enhancing the robustness of the noise translation network. In contrast, in inference stage, noisy input images are given directly to the noise translation network without adding extra Gaussian noise, because the direct noise augmentation to the input images degrades output quality. To establish a consistent Gaussian noise prior in both the training and inference stage, we inject Gaussian noise into every intermediate block of the noise translation network. Since the noise translation network is designed based on U-Net with residual connections between the input $I$ and the output $I_{\mathcal{T}}$, the distortion of the signal caused by injected Gaussian noise is alleviated, while allowing the noise translation network to utilize the Gaussian prior for transforming unseen real noise. Our ablation studies further demonstrate that the proposed Gaussian noise injection is crucial for the noise translation network to effectively translate unseen noise into Gaussian noise during the inference stage.

## 4 EXPERIMENTS

We demonstrate the effectiveness of the proposed approach on various benchmarks, evaluating performance on both in-distribution and out-of-distribution datasets. This section also provides an in-depth analysis of our algorithm, including detailed ablation studies and qualitative assessments.

Table 1: Quantitative comparison between other state-of-the-art real-world denoising networks and our adaptation framework-applied networks on the SIDD validation set (in-distribution) and other real-world benchmarks (out-of-distribution). We present the performance in terms of PSNR↑ (dB) and SSIM↑. Networks marked with asterisk (*) are evaluated using official out-of-the-box models.

| Architecture | Metric | In-distribution | Out-of-distribution | | | | | | | | |
|---|---|---|---|---|---|---|---|---|---|---|---|
| | | SIDD | Poly | CC | HighISO | iPhone | Huawei | OPPO | Sony | Xiaomi | OOD Avg. |
| MIRNet-v2* | PSNR | 39.76 | 37.39 | 35.93 | 38.15 | 40.41 | 38.06 | 39.62 | 43.89 | 35.39 | 38.60 |
| | SSIM | 0.9589 | 0.9798 | 0.9796 | 0.9774 | 0.9781 | 0.9681 | 0.9792 | 0.9893 | 0.9709 | 0.9778 |
| MPRNet* | PSNR | 39.63 | 37.47 | 35.92 | 38.00 | 40.13 | 38.29 | 39.70 | 43.88 | 35.46 | 38.61 |
| | SSIM | 0.9581 | 0.9765 | 0.9765 | 0.9728 | 0.9736 | 0.9668 | 0.9783 | 0.9889 | 0.9693 | 0.9753 |
| Uformer* | PSNR | 39.80 | 37.44 | 36.00 | 38.10 | 40.23 | 38.31 | 39.62 | 43.77 | 35.48 | 38.62 |
| | SSIM | 0.9590 | 0.9790 | 0.9792 | 0.9759 | 0.9743 | 0.9680 | 0.9784 | 0.9882 | 0.9708 | 0.9767 |
| Restormer* | PSNR | 39.93 | 37.63 | 36.31 | 38.24 | 40.05 | 38.36 | 39.49 | 44.02 | 35.62 | 38.85 |
| | SSIM | 0.9598 | 0.9790 | 0.9805 | 0.9753 | 0.9727 | 0.9671 | 0.9768 | **0.9889** | 0.9706 | 0.9745 |
| Restormer-ours | PSNR | 39.08 | **38.74** | **37.60** | **40.06** | **41.62** | **39.68** | **40.55** | **44.12** | **36.14** | **39.81** |
| | SSIM | 0.9558 | **0.9846** | **0.9861** | **0.9851** | **0.9751** | **0.9761** | **0.9794** | 0.9849 | **0.9747** | **0.9807** |
| NAFNet* | PSNR | 40.21 | 36.04 | 34.39 | 37.88 | 36.53 | 36.13 | 39.32 | 40.45 | 34.82 | 37.31 |
| | SSIM | 0.9609 | 0.9615 | 0.9784 | 0.9769 | 0.8896 | 0.9385 | 0.9764 | 0.9339 | 0.9657 | 0.9535 |
| NAFNet-ours | PSNR | 39.17 | **38.67** | **37.82** | **39.94** | **41.94** | **39.74** | **40.45** | **44.17** | **36.14** | **39.86** |
| | SSIM | 0.9566 | **0.9851** | **0.9876** | **0.9853** | **0.9805** | **0.9778** | **0.9796** | **0.9869** | **0.9745** | **0.9822** |
| KBNet* | PSNR | 40.26 | 36.79 | 35.21 | 38.05 | 37.93 | 35.14 | 37.73 | 41.65 | 34.23 | 37.09 |
| | SSIM | 0.9618 | 0.9785 | 0.9808 | 0.9785 | 0.9526 | 0.9459 | 0.9657 | 0.9784 | 0.9640 | 0.9681 |
| KBNet-ours | PSNR | 39.06 | **38.57** | **37.59** | **39.83** | **41.63** | **39.71** | **40.46** | **44.04** | **36.04** | **39.73** |
| | SSIM | 0.9559 | **0.9840** | **0.9859** | **0.9845** | **0.9752** | **0.9773** | **0.9794** | **0.9839** | **0.9739** | **0.9805** |

## 4.1 EXPERIMENTAL SETTINGS

**Training details** Since our approach is model-agnostic, we employ existing architectures such as NAFNet (Chen et al., 2022), Restormer (Zamir et al., 2022a), and KBNet (Zhang et al., 2023) as our image denoising network, which is pretrained on BSD400 (Martin et al., 2001), WED (Ma et al., 2016), and SIDD medium (Abdelhamed et al., 2018) datasets. BSD400 and WED datasets consist of clean images only, while SIDD dataset is composed of real noisy-clean image pairs. During training a denoising network, noisy images are generated by adding Gaussian noise with a standard deviation of 15 to clean images of BSD400 and WED datasets, and noisy images of SIDD datasets. Each training batch consists of images drawn equally from two sources: half from the combined BSD400 and WED datasets, and the other half from the SIDD dataset. The denoising models are trained for 200K iterations with a batch size of 32, except for Restormer, where the batch size is reduced to 4 due to the limitation of computational resources. After training a denoising network, we train the noise translation network with the SIDD dataset only, where noisy images are augmented by adding stochastic level of Gaussian noise with a range of 0 to 15. The noise translation network is trained for 5K iterations with a batch size of 4. Both the image denoising network and noise translation network adopt the AdamW (Loshchilov & Hutter, 2019) optimizer with an initial learning rate of $10^{-3}$, which is reduced using a cosine annealing schedule, down to $10^{-7}$ and $10^{-5}$, respectively. Each image is randomly cropped to $256 \times 256$ for training. All trainings were conducted using two NVIDIA RTX A6000 GPUs.

**Evaluation** To evaluate the generalization performance of our framework, we employ various real-world image denoising benchmarks. We conduct experiments using SIDD validation dataset (Abdelhamed et al., 2018), Poly (Xu et al., 2018), CC (Nam et al., 2016), HighISO (Yue et al., 2019), iPhone, Huawei, Oppo, Sony, and Xiaomi (Kong et al., 2023). The SIDD validation dataset consists of images with a resolution of $256 \times 256$ pixels, while the Poly, CC, and HighISO datasets contain images with a resolution of $512 \times 512$ pixels. Images from iPhone, Huawei, Oppo, Sony, and Xiaomi are $1024 \times 1024$ pixels in size.

Table 2: Quantitative results based on variations in the noisy input to the pretrained denoising network on the SIDD validation set (in-distribution) and other real-world benchmarks (out-of-distribution). $I$, $I + \mathcal{N}_5$, $I + \mathcal{N}_{10}$, and $I + \mathcal{N}_{15}$ represent the noisy input images with additional Gaussian noise levels of 0, 5, 10, and 15, respectively, which are fed into the pretrained Gaussian denoising network. We present performance in terms of PSNR↑ (dB) and SSIM↑.

| Input | Metric | In-distribution | Out-of-distribution | | | | | | | | |
|---|---|---|---|---|---|---|---|---|---|---|---|
| | | SIDD | Poly | CC | HighISO | iPhone | Huawei | OPPO | Sony | Xiaomi | OOD Avg. |
| $I$ | PSNR | 37.77 | 15.24 | 33.76 | 21.18 | 40.13 | 8.68 | 8.45 | 6.35 | 9.33 | 17.89 |
| | SSIM | 0.9360 | 0.3466 | 0.9139 | 0.5232 | 0.9734 | 0.1218 | 0.1138 | 0.0245 | 0.1845 | 0.4002 |
| $I + \mathcal{N}_5$ | PSNR | 38.15 | 27.07 | 34.97 | 32.30 | 15.28 | 16.99 | 13.79 | 12.82 | 14.96 | 22.93 |
| | SSIM | 0.9436 | 0.7010 | 0.9211 | 0.8227 | 0.2392 | 0.3943 | 0.2564 | 0.1912 | 0.3540 | 0.5359 |
| $I + \mathcal{N}_{10}$ | PSNR | 38.76 | 38.27 | 37.33 | 39.40 | 40.95 | 39.44 | 39.98 | 42.96 | 35.91 | 39.22 |
| | SSIM | 0.9536 | 0.9795 | 0.9850 | 0.9825 | 0.9638 | 0.9762 | 0.9768 | 0.9758 | 0.9728 | 0.9740 |
| $I + \mathcal{N}_{15}$ | PSNR | 39.16 | 38.08 | 36.26 | 38.85 | 41.12 | 38.71 | 39.69 | 43.42 | 35.25 | 38.95 |
| | SSIM | 0.9565 | 0.9834 | 0.9829 | 0.9808 | **0.9811** | 0.9719 | 0.9770 | **0.9886** | 0.9680 | 0.9767 |
| $I_{\mathcal{T}}$ | PSNR | 39.17 | **38.67** | **37.82** | **39.94** | **41.94** | **39.74** | **40.45** | **44.17** | **36.14** | **39.86** |
| | SSIM | 0.9566 | **0.9851** | **0.9876** | **0.9853** | 0.9805 | **0.9778** | **0.9796** | 0.9869 | **0.9745** | **0.9822** |

## 4.2 RESULTS AND ANALYSIS

**Denoising performance on real noise** Table 1 illustrates the performance of the proposed approach applied to the denoising networks Restormer, NAFNet and KBNet, along with the results from well-known real-world image denoising networks, including MPRNet (Zamir et al., 2021), MIRNet-v2 (Zamir et al., 2022b), and Uformer (Wang et al., 2022). For evaluating existing methods, we use the officially published models trained on the SIDD dataset. Our approach utilizes additional clean images for training the image denoising network. As shown in Table 1, incorporating our noise translation framework, denoted by Restormer-ours, NAFNet-ours and KBNet-ours, results in significantly improved PSNR and SSIM in most out-of-distribution (OOD) scenarios. '

**Comparisons with simple Gaussian noise addition** We validate the effectiveness of our noise translation network by comparing it to simply adding Gaussian noise. As shown in table 2, simply adding Gaussian noise to the input can result in fairly good generalization performance. However, some datasets perform better with the input $I + \mathcal{N}_{10}$, while others perform better with the input $I + \mathcal{N}_{15}$. This suggests that each image or dataset has different noise characteristics, necessitating a more flexible approach than merely adding a fixed level of Gaussian noise. Our noise translation network optimally transforms the input noise of each image into ideal Gaussian noise, leading to significant performance improvements across all datasets.

**Analysis of translated noise** Figure 3 visualizes noise component before and after our noise translation process. Real noise exhibits strong spatial correlation and signal dependency. These correlations are alleviated through the noise translation, transforming the noise to resemble ideal Gaussian noise. Figure 4 presents the analysis of the noise distribution using histograms. In the spatial domain, Gaussian noise follow a normal distribution, while in the frequency domain, it follows Rayleigh distribution, as mentioned in Section 3.2.2. The original real-noise distribution significantly deviates from the expected target: Gaussian distribution in spatial domain and Rayleigh distribution in frequency domain. After the translation, the noise closely follows the target distributions, demonstrating the effectiveness of our method in aligning the noise characteristics with the ideal Gaussian noise in both domains. This indicates that our method successfully transforms the noise into spatially uncorrelated, *i.i.d* Gaussian noise.

**Qualitative results** Figure 5 shows qualitative results of the SIDD validation dataset. Although our method appears to compromise on in-distribution performance, this is due to the severe overfitting of other models, which even reconstruct the unnecessary artifacts prevalent in the training set. In contrast, our method preserves visual quality without overfitting, effectively removing noise without introducing artifacts. Figure 6 presents qualitative results of denoising models applied to

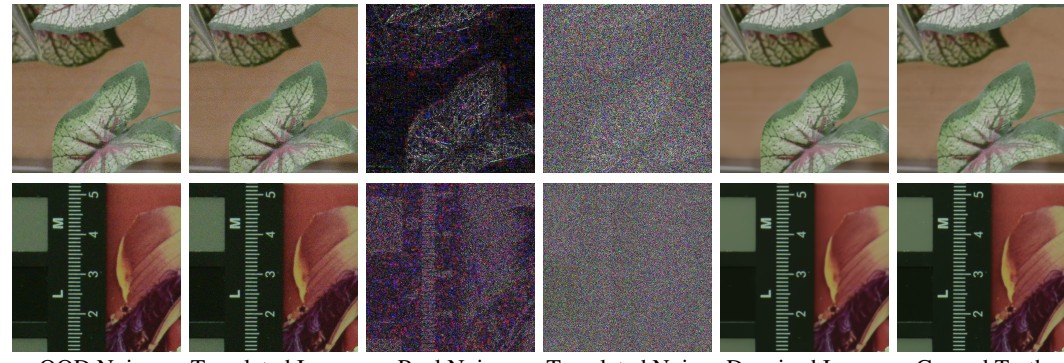

| OOD Noisy | Translated Image | Real Noise | Translated Noise | Denoised Image | Ground Truth |

Figure 3: Visualization of noise translation and denoised results. The noisy input image in the top row is from the Poly dataset, while the one in the bottom row is from the CC dataset. The original real noise exhibits strong signal dependency, whereas the translated noise closely resembles Gaussian noise, leading to improved denoising performance. For better visualization, the noise is shown as the absolute value scaled by a factor of 10.

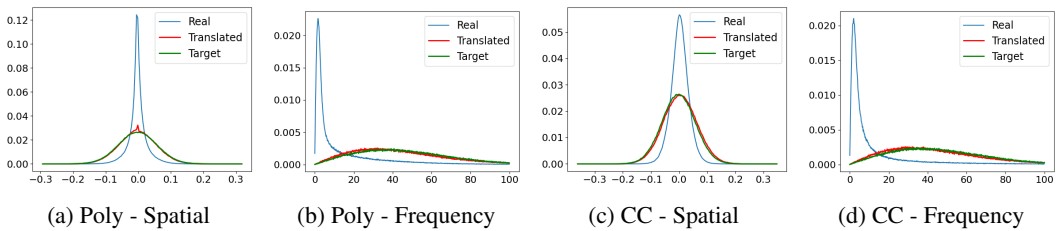

|   (a) Poly - Spatial   |   (b) Poly - Frequency   |   (c) CC - Spatial   |   (d) CC - Frequency   |

Figure 4: Histogram of noise distribution in both spatial and frequency domains. The real-noise distributions of left two plots and right two plots are each obtained from single images in the Poly and CC datasets, respectively. Real and translated noise distributions are obtained by subtracting the ground truth image from the input noisy image or the corresponding translated image. Target noise corresponds to the Gaussian noise with a level of 15, which the denoising network has been pretrained to remove. The original real noise is shown in blue, the translated noise is shown in red, and target noise is shown in green.

various real-world OOD datasets, where our method significantly outperforms other denoising models. Additional qualitative results on various benchmarks are provided in Section B.4.

## 4.3 ABLATIVE RESULTS

**Impact of Gaussian noise injection and explicit noise translation loss**    Table 3 illustrates the performance gains attributed to each component of the proposed method. The baseline translation in Table 3 refers to the results obtained by applying our method only with the implicit noise translation loss without Gaussian noise injection and explicit noise translation loss. When Gaussian noise injection is applied, there is a significant improvement in out-of-distribution (OOD) performance. Lastly, by incorporating the explicit noise translation loss, we observe the best performance gains.

**Effects of hyperparameters**    Table 4 presents the ablative results of our hyperparameters, including pretraining noise level ($\sigma$), noise injection level ($\tilde{\sigma}$), explicit loss weight ($\alpha$), and spatial frequency ratio ($\beta$). The pretraining level part of Table 4 shows the ablation results for the Gaussian noise levels added to create noisy input during denoising network pretraining. As the noise level increased, the performance on in-distribution (ID) consistently decreased. For out-of-distribution (OOD), the performance improved until noise level of 15, beyond which it began to degrade, due to oversmoothing effects caused by learning to handle strong noise. The noise injection level part presents the ablation results for Gaussian noise injection levels. Increasing the noise level led to a decline in ID performance, while OOD performance improved up to a certain point. The explicit loss weight and spatial-frequency ratio parts show the ablation results for the $\alpha$ and $\beta$ values in Eqs. (12)

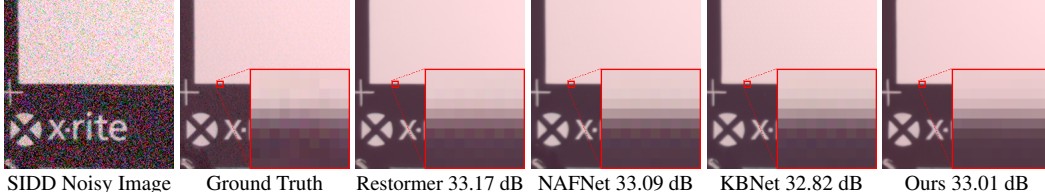

| SIDD Noisy Image | Ground Truth | Restormer 33.17 dB | NAFNet 33.09 dB | KBNet 32.82 dB | Ours 33.01 dB |

Figure 5: Denoised results of an in-distribution noisy image. Although all methods demonstrate reasonable performance, only our approach avoids overfitting, thereby preventing the zipper artifact in the ground truth images from the training set. Zoom in for better comparison.

| MIRNet 40.38 dB | MPRNet 40.48 dB | Uformer 40.51 dB | Ground Truth |

| Poly Noisy | Restormer 40.72 dB | NAFNet 39.97 dB | KBNet 40.45 dB | **Ours 43.59 dB** |

| MIRNet 38.38 dB | MPRNet 36.69 dB | Uformer 37.48 dB | Ground Truth |

| CC Noisy | Restormer 37.40 dB | NAFNet 36.99 dB | KBNet 38.28 dB | **Ours 42.18 dB** |

| MIRNet 34.70 dB | MPRNet 34.26 dB | Uformer 34.58 dB | Ground Truth |

| HighISO Noisy | Restormer 34.59 dB | NAFNet 35.60 dB | KBNet 35.53 dB | **Ours 36.44 dB** |

Figure 6: Comparison between the qualitative results of various denoising networks including ours (noise translation network with pretrained NAFNet), on the out-of-distribution (OOD) datasets. Our result displays cleaner outputs compared to other state-of-the-art networks that are directly trained from a real-noise dataset.

Table 3: Effects of using Gaussian noise injection and explicit loss. We present denoising performance in terms of PSNR↑ (dB) and SSIM↑.

| | | In-distribution | Out-of-distribution | | | | | | | | |
|---|---|---|---|---|---|---|---|---|---|---|---|
| | Metric | SIDD | Poly | CC | HighISO | iPhone | Huawei | OPPO | Sony | Xiaomi | OOD Avg. |
| Baseline Translation | PSNR | 39.35 | 38.32 | 37.25 | 39.22 | 40.80 | 39.24 | 39.75 | 43.86 | 35.74 | 39.27 |
| | SSIM | 0.9573 | 0.9820 | 0.9864 | 0.9794 | 0.9700 | 0.9727 | 0.9745 | 0.9857 | 0.9683 | 0.9774 |
| + Gaussian Injection | PSNR | 39.05 | 38.54 | 37.58 | 39.79 | 41.53 | 39.68 | 40.40 | 43.89 | 36.00 | 39.61 |
| | SSIM | 0.9556 | 0.9835 | 0.9866 | 0.9844 | 0.9737 | 0.9773 | 0.9790 | 0.9827 | 0.9737 | 0.9801 |
| + Explicit Loss | PSNR | 39.37 | 37.91 | 37.16 | 39.07 | 40.28 | 37.85 | 39.06 | 42.72 | 35.21 | 38.66 |
| | SSIM | 0.9574 | 0.9791 | 0.9862 | 0.9783 | 0.9650 | 0.9613 | 0.9670 | 0.9789 | 0.9605 | 0.9720 |
| + Both | PSNR | 39.17 | **38.67** | **37.82** | **39.94** | **41.94** | **39.74** | **40.45** | **44.17** | **36.14** | **39.86** |
| | SSIM | 0.9566 | **0.9851** | **0.9876** | **0.9853** | **0.9805** | **0.9778** | **0.9796** | **0.9869** | **0.9745** | **0.9822** |

Table 4: Sensitivity results on various hyperparameters in our framework.

| Pretraining Level | | | Noise Injection Level | | | Explicit Loss Weight | | | Spatial-Frequency Ratio | | |
|---|---|---|---|---|---|---|---|---|---|---|---|
| $\sigma$ | SIDD | OOD Avg. | $\tilde{\sigma}$ | SIDD | OOD Avg. | $\alpha$ | SIDD | OOD Avg. | $\beta$ | SIDD | OOD Avg. |
| 5 | 39.37 | 38.96 | 0 | 39.37 | 38.66 | 0 | 39.05 | 39.61 | 0 | 39.10 | 39.78 |
| 10 | 39.29 | 39.64 | 1 | 39.33 | 39.24 | 0.001 | 39.03 | 39.65 | 0.001 | 39.15 | 39.82 |
| 15 | 39.17 | **39.86** | 5 | 39.17 | 39.66 | 0.005 | 39.07 | 39.72 | 0.002 | **39.17** | **39.86** |
| 20 | 38.88 | 39.53 | 15 | 39.08 | 39.77 | 0.01 | 39.09 | 39.76 | 0.005 | 39.07 | 39.82 |
| 25 | 38.73 | 39.21 | 50 | 39.11 | 39.80 | 0.05 | **39.17** | **39.86** | 0.01 | 39.07 | 39.56 |
| | | | 100 | 39.17 | **39.86** | 0.1 | 39.05 | 39.67 | 0.02 | 39.07 | 38.97 |
| | | | 200 | 39.05 | 39.70 | 0.5 | 38.90 | 37.80 | 0.05 | 38.57 | 37.41 |

and (11), respectively. Overall, the ablation experiments determined the optimal hyperparameters as follows: a pretraining noise level $\sigma = 15$, a Gaussian noise injection level $\tilde{\sigma} = 100$, $\alpha = 5 \times 10^{-2}$, and $\beta = 2 \times 10^{-3}$.

## 5 CONCLUSION

We presented a novel noise translation framework for robust image denoising. Our framework allows us to effectively remove various unseen real noise, even with limited amount of training data. By employing the noise translation network, we transform arbitrary out-of-distribution (OOD) noise into Gaussian noise for which our image denoising network has learned during training. The noise translation network is designed with well-motivated loss functions and architecture, enabling effective noise translation while preserving image contents. Our experiments demonstrate that the proposed approach significantly outperforms state-of-the-art denoising models in diverse OOD real-noise benchmarks. Finally, we highlight that the generalization issue remains a critical challenge in image denoising, and our approach offers a promising solution to address this problem.

## 6 REPRODUCIBILITY STATEMENT

To ensure reproducibility of our work, we have made efforts to include key details in the main text and appendix. Furthermore, we provide the complete source code in the supplementary material to facilitate easy reproduction of the experiments and results.

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

# A  PROOF ON WASSERSTEIN DISTANCE

Let $P$ and $Q$ represent two probability distributions over $\mathbb{R}^d$. We use $X \sim P$ and $Y \sim Q$ to denote random variables with the distributions $P$ and $Q$, respectively. The $p$-Wasserstein distance between two probability measures $P$ and $Q$ is defined as follows:

$$W_p(P,Q) = \left( \inf_{J \in \mathcal{J}(P,Q)} \int \|x - y\|^p dJ(x,y) \right)^{1/p},  \tag{13}$$

where $\mathcal{J}(P,Q)$ is the set of all joint distributions (or couplings) $J$ on $(X,Y)$ that have marginals $P$ and $Q$. This formulation describes the minimum cost of transporting mass from distribution $P$ to distribution $Q$ using the coupling $J$, with the cost measured as the $p$-th power of the distance between points $x$ and $y$.

In the Monge formulation, the goal is to find a transport map $T : \mathbb{R}^d \to \mathbb{R}^d$ such that the push-forward of $P$ under $T$, denoted as $T_{\#}P$, equals $Q$. This problem can be mathematically formulated as:

$$\inf_T \int |x - T(x)|^p dP(x),  \tag{14}$$

where the map $T$ moves the distribution $P$ to $Q$. However, an optimal map $T$ may not always exist. In such cases, the Kantorovich formulation is used, allowing mass at each point to be split and transported to multiple locations, leading to a coupling-based approach.

For the specific case of $p = 1$, known as the Earth Mover's Distance, the dual formulation of the Wasserstein distance can be expressed as:

$$W_1(P,Q) = \sup_{f \in F} \left( \int f(x)dP(x) - \int f(x)dQ(x) \right),  \tag{15}$$

where $F$ represents the set of all Lipschitz continuous functions $f : \mathbb{R}^d \to \mathbb{R}$ such that $|f(y) - f(x)| \le \|x - y\|$ for all $x, y \in \mathbb{R}^d$. Then, the 1-Wasserstein distance is given by:

$$W_1(P,Q) = \int_0^1 |F^{-1}(z) - G^{-1}(z)|dz,  \tag{16}$$

where $F^{-1}$ and $G^{-1}$ denote the quantile functions (inverse CDFs) of $P$ and $Q$, respectively.

When $P$ and $Q$ are empirical distributions based on the datasets, $X_1, X_2, \ldots, X_n$ and $Y_1, Y_2, \ldots, Y_n$, each of size $n$, the Wasserstein distance can be computed as a function of the order statistics:

$$W_1(P,Q) = \sum_{i=1}^{n} |X_{(i)} - Y_{(i)}|,  \tag{17}$$

where $X_{(i)}$ and $Y_{(i)}$ denote the $i$-th order statistics of the datasets $X_1, X_2, \ldots, X_n$ and $Y_1, Y_2, \ldots, Y_n$.

In our approach, we utilize (17) to formulate $\mathcal{L}_{\text{spatial}}$ in (6) and $\mathcal{L}_{\text{freq}}$ in (10), which are employed during training to explicitly transport the translated noise distribution towards the target Gaussian distribution. We refer to (Villani & Society, 2003) for a detailed discussion on Wasserstein distances and optimal transport.

# B  ADDITIONAL EXPERIMENTAL RESULTS

## B.1  TRAINING STABILITY

To assess the stability of training our noise translation network, we conducted five independent training runs using different random seeds and evaluated the PSNR metrics across multiple datasets. The standard deviations for each dataset were as follows: SIDD (0.023), Poly (0.007), CC (0.021), High-ISO (0.009), iPhone (0.008), Huawei (0.010), OPPO (0.016), Sony (0.022), and Xiaomi (0.020). The average standard deviation across all datasets was 0.013, validating the stability of our method. In this paper, we report the results obtained with random seed 8.

Table 5: Comparison of parameter counts and MACs for denoising networks and our noise translation network. We report the number of parameters and MACs at inference, estimated with an input size of 256×256.

| Architecture | Parameters (M) | MACs (G) |
|---|---|---|
| MIRNet-v2 | 5.86 | 140.34 |
| MPRNet | 15.74 | 588.14 |
| Uformer | 50.9 | 89.5 |
| Restormer | 26.1 | 141.0 |
| NAFNet | 115.86 | 63.6 |
| KBNet | 104.93 | 58.19 |
| Noise Translation Network | 0.29 | 1.07 |

Table 6: Quantitative comparisons between various state-of-the-art image denoising methods on the SIDD validation set and multiple real-noise benchmarks. We present the results in terms of PSNR↑ (dB) and SSIM↑. The table includes both supervised and self-supervised denoising methods. Networks marked with an asterisk (*) are evaluated using official out-of-the-box models.

| | Architecture | Metric | SIDD | Poly | CC | HighISO | iPhone | Huawei | OPPO | Sony | Xiaomi | Total Avg. |
|---|---|---|---|---|---|---|---|---|---|---|---|---|
| Others | MaskDenoising* | PSNR | 28.66 | 34.56 | 33.87 | 34.61 | 36.54 | 34.89 | 35.30 | 37.89 | 33.46 | 34.20 |
| | | SSIM | 0.7127 | 0.9553 | 0.9703 | 0.9649 | 0.9273 | 0.9586 | 0.9593 | 0.9354 | 0.9531 | 0.9263 |
| | CLIPDenoising* | PSNR | 34.79 | 37.54 | 36.30 | 38.01 | 40.09 | 38.74 | 39.56 | 42.94 | 35.50 | 38.39 |
| | | SSIM | 0.8982 | 0.9794 | 0.9809 | 0.9771 | 0.9685 | 0.9715 | 0.9769 | 0.9824 | 0.9707 | 0.9672 |
| | DnCNN-AFM* | PSNR | 38.29 | 37.71 | 36.81 | 39.12 | 40.56 | 38.33 | 40.13 | **44.66** | 35.25 | 38.54 |
| | | SSIM | 0.9474 | 0.9800 | 0.9828 | 0.9797 | 0.9769 | 0.9679 | 0.9795 | **0.9901** | 0.9665 | 0.9745 |
| Self-supervised | R2R* | PSNR | 35.06 | 36.81 | 35.26 | 37.33 | 39.19 | 38.29 | 39.32 | 41.46 | 35.36 | 37.12 |
| | | SSIM | 0.9150 | 0.9722 | 0.9756 | 0.9712 | 0.9606 | 0.9663 | 0.9739 | 0.9729 | 0.9664 | 0.9638 |
| | AP-BSN* | PSNR | 36.32 | 35.88 | 33.13 | 36.66 | 39.82 | 37.01 | 39.04 | 40.04 | 33.37 | 36.70 |
| | | SSIM | 0.9281 | 0.9751 | 0.9732 | 0.9777 | 0.9766 | 0.9628 | 0.9746 | 0.9798 | 0.9548 | 0.9669 |
| | SSID* | PSNR | 37.39 | 37.13 | 34.93 | 38.24 | 40.85 | 37.22 | 39.10 | 42.89 | 34.25 | 38.00 |
| | | SSIM | 0.9338 | 0.9799 | 0.9805 | 0.9808 | **0.9816** | 0.9652 | 0.9738 | 0.9878 | 0.9562 | 0.9700 |
| Supervised | NAFNet-ours | PSNR | **39.17** | **38.67** | **37.82** | **39.94** | **41.94** | **39.74** | **40.45** | 44.17 | **36.14** | **39.67** |
| | | SSIM | **0.9566** | **0.9851** | **0.9876** | **0.9853** | 0.9805 | **0.9778** | **0.9796** | 0.9869 | **0.9745** | **0.9793** |

## B.2 PARAMETERS AND MACS

Table 5 presents a comparison of our noise translation network with other image denoising networks in terms of the number of parameters and multiply–accumulate operations (MACs). Our noise translation network is significantly smaller in size compared to the image denoising networks, resulting in negligible additional computational cost during inference.

## B.3 COMPARISON WITH SELF-SUPERVISED AND GENERALIZATION METHODS

Table 6 further presents the performance of denoising models trained using other generalization methods and self-supervised approaches on real-world noise datasets. All models are evaluated using publicly available official weights. For MaskDenoising (Chen et al., 2023), which is trained solely with Gaussian noise ($\sigma = 15$), the performance on real-world noise datasets is notably low. In the case of CLIPDenoising (Cheng et al., 2024), it does not utilize real noise during training but instead relied on synthetic noise generated with Poisson-Gaussian models for sRGB denoising. As a result, its average performance on real-world datasets remains quite poor. DnCNN-AFM (Ryou et al., 2024) is trained in a supervised manner on the real noise dataset (SIDD), while also employing an adversarial noise generation strategy to increase robustness. Although it performs better than previous methods, its performance still falls short compared to our proposed approach.

Additionally, self supervised methods R2R (Pang et al., 2021), AP-BSN (Lee et al., 2022) and SSID (Li et al., 2023), listed in Table 6 are trained using only the noisy images from the real-noise dataset SIDD (Abdelhamed et al., 2018). As a result, their performance is consistently lower compared to our supervised method across all datasets.

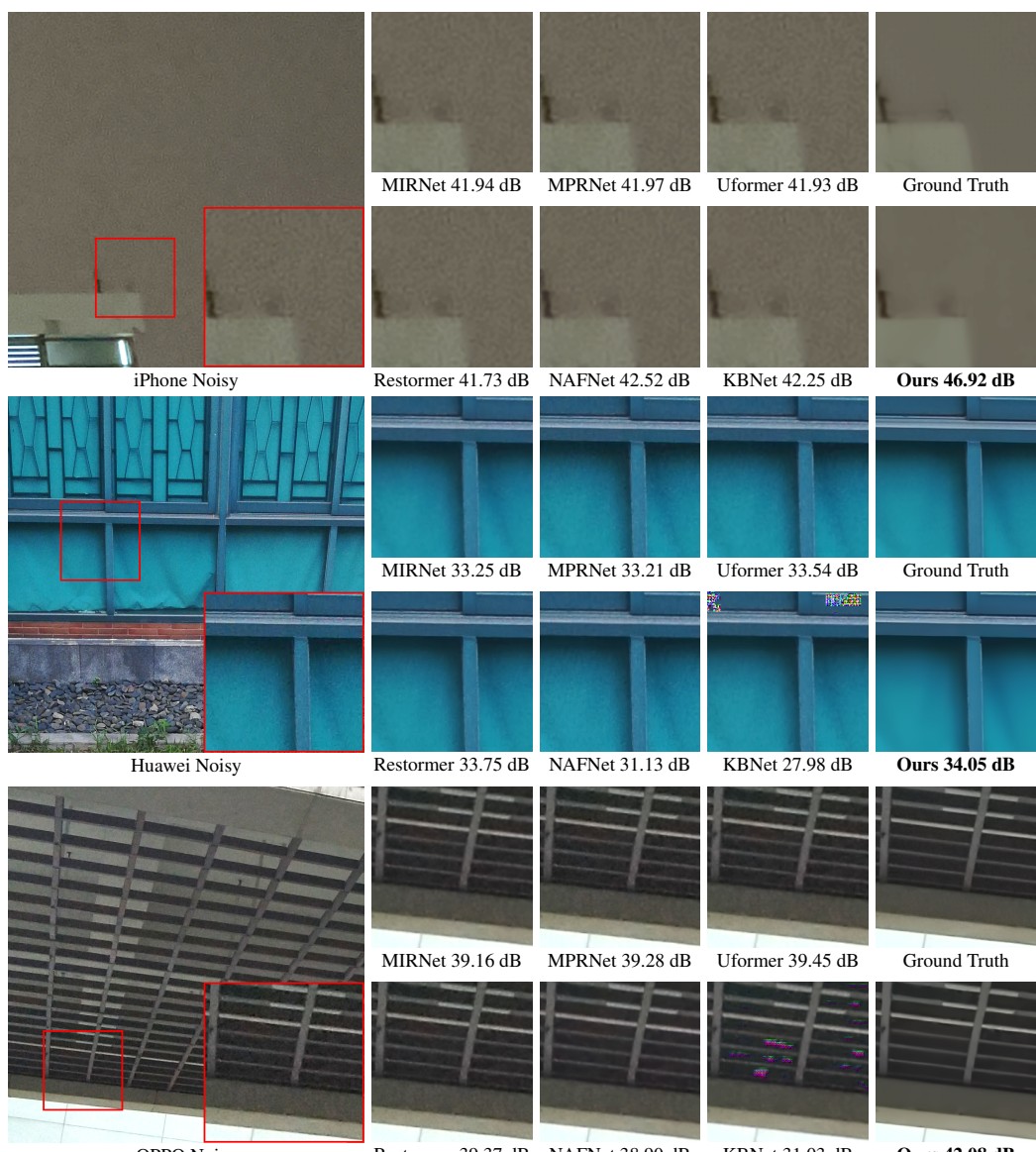

Figure 7: Additional qualitative results on iPhone, Huawei, and OPPO dataset.

## B.4 QUALITATIVE RESULTS

Extensive qualitative results on all datasets are shown in Figures 7 and 8. Our method significantly surpasses the Peak Signal-to-Noise Ratio (PSNR) scores of other denoising models on all OOD datasets (iPhone, Huawei, OPPO, Sony, Xiaomi). Notably, the output images of KBNet (Zhang et al., 2023), exhibit visible breakdowns due to severe overfitting to the training set. For the in-distribution SIDD dataset, only our method produces clean results without generating unnecessary zipper artifacts. Additionally, we include the denoised results of photos captured with our Galaxy S22+ smartphone in Figure 9.

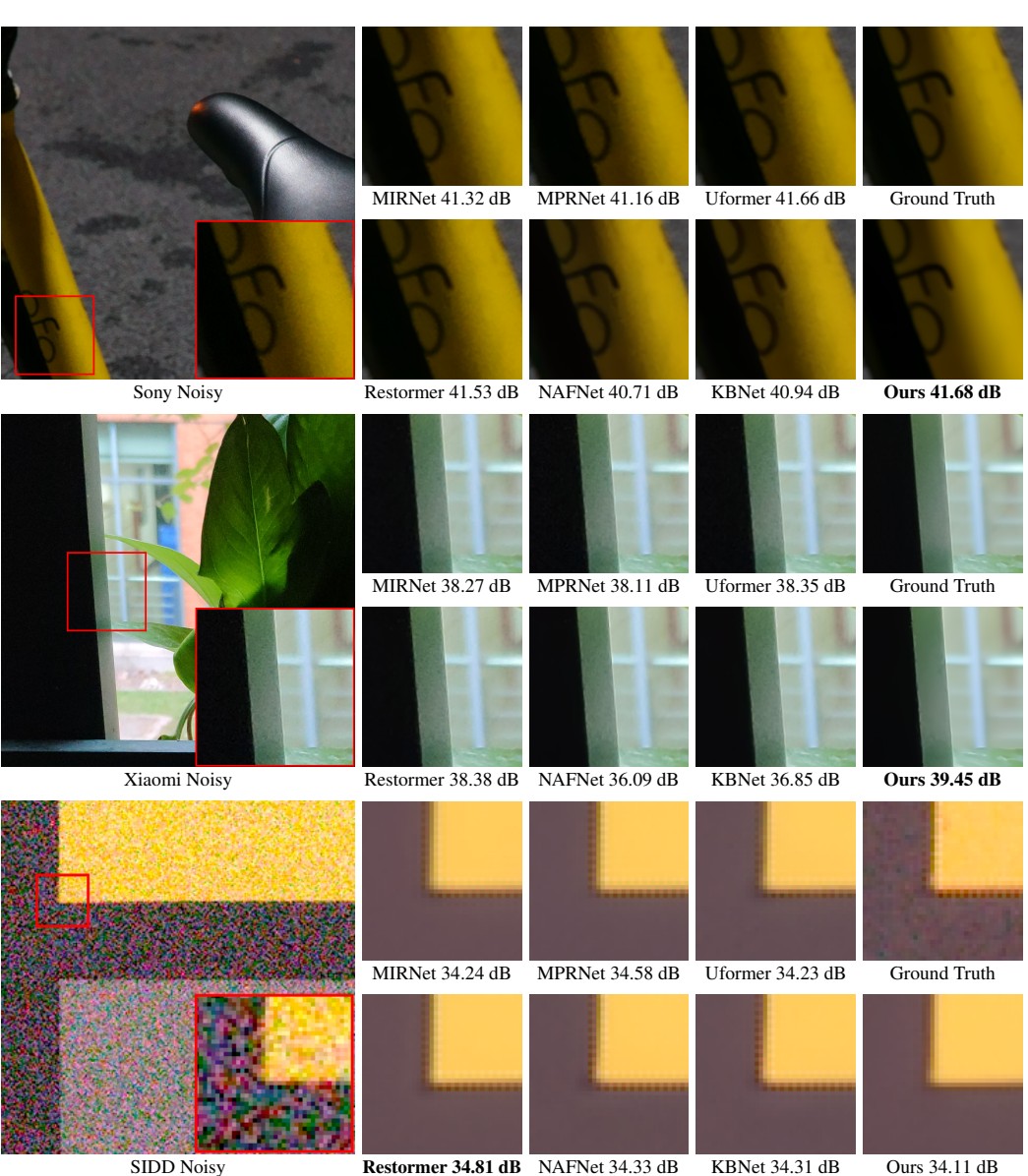

Figure 8: Additional qualitative results on Sony, Xiaomi, and SIDD datasets.

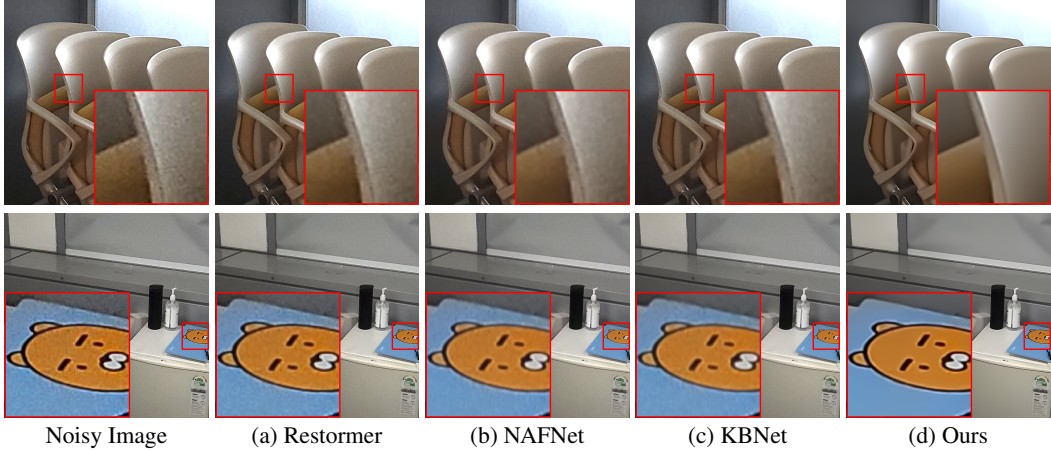

| Noisy Image | (a) Restormer | (b) NAFNet | (c) KBNet | (d) Ours |

Figure 9: Denoised results of images captured by our Galaxy S22+ smartphone. Unlike existing denoising models, our approach effectively removes challenging real-world noise.

