# OpenReview forum: "Learning to Translate Noise for Robust Image Denoising"
_ICLR.cc/2025/Conference — ICLR 2025 Conference Withdrawn Submission_

### Official Review · Reviewer_k3tR · 2024-10-19

**Soundness:** 3
**Presentation:** 3
**Contribution:** 3
**Rating:** 6
**Confidence:** 4

**Summary:**

This paper proposes a noise translation network designed for robust image denoising. The proposed approach involves using a noise translation network to convert real-world noise into Gaussian noise, followed by the application of a pretrained denoising network to remove the noise. The main contribution of this work lies in the development of a training method for the noise translation network, which incorporates both implicit and explicit noise translation losses. The authors validate the performance of the proposed method across various datasets and different denoising networks. The method demonstrates improved out-of-distribution denoising performance compared to the original denoising networks.

**Strengths:**

1. This paper is well-written and easy to follow.
2. The method is validated across various datasets and demonstrates strong performance on out-of-distribution data.
3. The training strategy for the noise translation network is novel, and the authors have conducted a comprehensive ablation analysis.

**Weaknesses:**

The methodology is not entirely clear to me. Based on the experiments, the noise translation network is trained on the SIDD dataset, suggesting that the model is designed to transfer noise from SIDD data to Gaussian noise. However, I am curious as to why the model is also able to transfer noise from other datasets to Gaussian noise. Does this imply that the noise distribution of the other datasets is very similar to that of the SIDD dataset? If the noise distribution of the other datasets differs significantly from the training dataset, would this method still be effective? I believe the authors should provide further clarification on this point.
Also, I find the performance decrease for in-distribution data to be quite substantial; it appears to drop by nearly 1 dB, which represents a significant gap.

**Questions:**

1. What are the limitations of this noise translation network? For instance, if the noise translation network is trained to transfer Poisson noise to Gaussian noise, does it still apply to other types of noise? Additionally, must the out-of-distribution noise belong to the same distribution family?
2. The denoising network is pretrained with both Gaussian noise and SIDD noise. Does this imply that the noise translation network may not perfectly transfer the noise to Gaussian noise? What would happen if the denoising network were only pretrained on Gaussian noise?

---

> ### Author Response · Authors · 2024-11-13
>
> Dear reviewer k3tR,
>
> Thank you for your detailed review and feedback on our paper. We have posted the rebuttal addressing each of your comments and concerns. We appreciate your constructive insights, which have guided us to enhance the clarity and depth of our paper.
>
> Thank you once again for your time and valuable feedback.
>
> Best regards, The Authors

---

### Official Review · Reviewer_jgKR · 2024-10-29

**Soundness:** 3
**Presentation:** 3
**Contribution:** 3
**Rating:** 5
**Confidence:** 5

**Summary:**

This work tackles the challenge of poor generalization in deep learning-based image denoising methods when handling real-world noise. The authors propose a novel approach that first translates complex input noise into well-known Gaussian noise, which is then removed using conventional denoising networks trained with additive white Gaussian noise (AWGN). This strategy leads to improved robustness and consistency in denoising.
The approach features a carefully designed architecture for noise translation along with specialized loss functions to reduce spatial correlations in the input noise, allowing it to follow a Gaussian distribution. Experimental results demonstrate that this method improves robustness and generalizability, outperforming state-of-the-art techniques across various benchmarks.

**Strengths:**

This work presents a novel framework that adapts real-world noise distributions into a well-understood Gaussian noise distribution, which is more easily removable by conventional denoising networks. To facilitate this noise adaptation, the authors propose novel loss functions that effectively control the distribution of the translated noise. Notably, this work provides extensive experimental results on image denoising benchmark datasets with diverse noise types, demonstrating that this intuitive approach significantly enhances the generalization performance of conventional denoising networks when applied to unknown real-world noise.

**Weaknesses:**

Improving the generalization performance of denoising networks and enhancing denoising quality on out-of-distribution (OOD) datasets is a critical challenge. To address this, the authors propose a dedicated adaptation module and novel loss functions. To the best of my knowledge, these proposed loss functions are new to the field and are thoughtfully designed to adapt input noise with an unknown distribution to a Gaussian distribution. However, as shown in Table 3, the performance gains achieved by these loss functions are modest compared to a simpler Gaussian injection approach, which somewhat limits the method’s superiority.
Additionally, I have concerns about the experimental setup in this work; please see the “Questions” section for further details.
Furthermore, a recent approach (Kim et al., “LAN: Learning to Adapt Noise for Image Denoising,” CVPR24) introduces a similar concept and architecture but trains the noise translation module in an unsupervised manner to convert unknown real noise to Gaussian noise, unlike this work, which relies on supervised training. Given this difference in reliance on supervision, a comparative analysis of the advantages and disadvantages relative to LAN, along with further insights, would be beneficial.

**Questions:**

1. To obtain the results in Tables 1-4, the denoising network trained on Gaussian noise is frozen, and only the translation module is exposed to real noise from the SIDD dataset for training. Then, this network is tested on datasets like Poly and CC, which are entirely different from SIDD, to validate its OOD performance. However, in real-world scenarios, when would such a setup occur? It raises concerns about whether this experimental setting might be somewhat unrealistic. How about exposing the translation module to Poisson or Gamma-Poisson noise when training it, then translating it to Gaussian and evaluating the OOD performance?

2. Additionally, while the architecture of the denoising network uses conventional models without modification, training these solely with Gaussian noise at a level of 15 seems to overly restrict their generalization performance. Considering that conventional Gaussian denoising networks are typically trained to flexibly handle Gaussian noise levels within the range [0–50], it would be beneficial in these experiments as well to first enhance the denoising network's generalization performance by simply exposing to more diverse Gaussian noise levels before conducting the OOD experiments.

3. The authors argue that the baseline model overfits to the SIDD dataset and, as a result, does not perform well on external datasets. In that case, wouldn’t it be possible to provide a comparison where dropout or other regularization techniques are applied during fine-tuning on SIDD to prevent overfitting?

---

> ### Author Response · Authors · 2024-11-13
>
> Dear reviewer jgKR,
>
> Thank you for your detailed review and feedback on our paper. We have posted the rebuttal addressing each of your comments and concerns. We appreciate your constructive insights, which have guided us to enhance the clarity and depth of our paper.
>
> Thank you once again for your time and valuable feedback.
>
> Best regards, The Authors

---

### Official Review · Reviewer_Nvpd · 2024-11-03

**Soundness:** 3
**Presentation:** 2
**Contribution:** 2
**Rating:** 5
**Confidence:** 4

**Summary:**

This paper addresses the issue of generalization in denoising networks. The authors propose a noise transfer mechanism that can convert any type of noise into Gaussian noise, enabling the use of Gaussian denoisers for real-world noise reduction. In their approach, they design two loss functions and a specialized structure for the noise transfer network.

Initially, the authors train the Gaussian denoiser using clean images and noisy images from the SIDD dataset. They then fix the weights of the denoiser and utilize the training data from the SIDD dataset to train the noise transfer network. The authors provide a comprehensive experimental analysis to demonstrate the effectiveness of the noise transfer network.

**Strengths:**

The performance after transfer is consistently better across different sensor noise types compared to before the transfer.
The loss function is designed from a novel perspective.

**Weaknesses:**

The comparison is somewhat unfair. The settings in the paper actually include both Gaussian image pairs and SIDD training data pairs. Therefore, a more appropriate baseline for comparison would be a denoiser trained simultaneously on both Gaussian noise and SIDD noise. With more training data and model parameters, it is likely that the denoiser could perform well even without the noise transfer network.

**Questions:**

1、Are the noise transfer network and the Gaussian denoiser paired? Can the noise transfer network trained on NAFNet be applied to Restormer? If the network indeed converts noise to Gaussian, then any Gaussian denoiser should be compatible with the same noise transfer network, which would be more practical from an application standpoint.

2、Why is the Gaussian noise score in Table 2 so poor?

3、During the training process, real noise training pairs from SIDD are still needed. However, the spatial correlation in SIDD is not very strong, typically around 2-4 pixels. For datasets with significant pattern noise, is there a need to collect new real noise pairs?

---

> ### Author Response · Authors · 2024-11-13
>
> Dear reviewer Nvpd,
>
> Thank you for your detailed review and feedback on our paper. We have posted the rebuttal addressing each of your comments and concerns. We appreciate your constructive insights, which have guided us to enhance the clarity and depth of our paper.
>
> Thank you once again for your time and valuable feedback.
>
> Best regards, The Authors

---

### Official Review · Reviewer_UMZZ · 2024-11-04

**Soundness:** 2
**Presentation:** 3
**Contribution:** 2
**Rating:** 5
**Confidence:** 3

**Summary:**

Existing denoising networks perform poorly on real-world noise, or noise that has a different distribution than that seen during training. This paper proposes a noise translation network that learns to transform the noise in an image into noise that is closer in distribution to that of a pre-trained denoising network. The latter is Gaussian noise. The proposed noise translation network consists of a set of computation blocks that inject Gaussian noise throughout the network. This network is trained with a pixel wise implicit loss and two other auxiliary losses.

**Strengths:**

This paper is well-written and it is clear from their reported results that the proposed method can enhance the image denoising process. Based on table in the appendix, this method also has order of magnitudes less parameter counts and MACs as compared to the other denoising networks. The figures are easy to understand and well designed, especially Figure 2.

**Weaknesses:**

It is surprising that the authors did not cite LAN [1] nor did the authors compare with LAN[1], despite the works being incredibly similar. This is one of the biggest weaknesses of this paper. The authors should address why this was not mentioned in their paper and should give a thorough comparison between their method and LAN. A quick glance at table 1 of this submission, row “Restormer-ours” PSNR on Poly is 38.74 dB while LAN[1] reports 39.23 dB, 39.30 dB,  39.28 using ZS-N2N. Even for the other supervised loss, the results are still higher than those of this submission (39.09 dB, 39.14dB, 39.17dB). LAN  exhibits notable performance improvement even when using only 5 iterations. Both works aim to translate or adapt noise prior to passing the adjusted image to a pretrained denoising network.

Second, the proposed noise translation network is ultimately trained with L_implicit which inherently means that this would be equivalent to adding additional parameters to the denoising network.

Third, the proposed translation network also has to be retrained for each denoising network. Have the authors considered a one time train adapter that can be applied to multiple denoising networks?


[1] Kim, Changjin, Tae Hyun Kim, and Sungyong Baik. "LAN: Learning to Adapt Noise for Image Denoising." Proceedings of the IEEE/CVF Conference on Computer Vision and Pattern Recognition. 2024.

**Questions:**

Line 213 “still insufficient to ensure that nT is spatially uncorrelated.” can you explain why this is desired?
The L_freq loss seems redundant and the ablations also demonstrate that when Beta=0, this loss has similar performance. Please explain more clearly the motivation for this loss that is not already captured in L_implicit. Can you also point to existing works that use either of these losses?

---

> ### Author Response · Authors · 2024-11-13
>
> Dear reviewer UMZZ,
>
> Thank you for your detailed review and feedback on our paper.
> We have posted the rebuttal addressing each of your comments and concerns.
> We appreciate your constructive insights, which have guided us to enhance the clarity and depth of our paper.
>
> Thank you once again for your time and valuable feedback.
>
> Best regards,
> The Authors

---

### Official Review · Reviewer_6pYh · 2024-11-08

**Soundness:** 3
**Presentation:** 3
**Contribution:** 2
**Rating:** 3
**Confidence:** 4

**Summary:**

This study proposes a new image denoising method, mainly aiming to improve its generalization performance to unseen practical complex noise. The main methodology is to build a noise translation network, purposing to translate diverse noises with unknown types into Gaussian, and thus make the image denoising pretrained on Gaussian noisy data reasonable to be used in such images.

**Strengths:**

The idea is clear, reasonable and interesting.
The experiments seem to be comprehensive, and the superiority of the proposed method is evident.

**Weaknesses:**

Although the proposed main methodology is reasonable, but I still have the following major concerns.

Firstly, two networks are involved in the proposed framework, including an image denoising network and a noise translation one. It seems that the two networks have been separately trained, and the former is pretrained and fixed in the training process of the latter one. Such a training manner tends to make the noise translation network's performance highly dependent on the capability of the former one. I think the two components should be more rationally trained simultaneously, to make them ameliorate each other's ability mutually. I highly recommend authors to read the following paper:
Dual Adversarial Network: Toward Real Noise Removal and Noise Generation. ECCV, 2020
which also investigated the real complex noise issue, but can learn the noise generation mechanism and denoiser together in a consistent and unified framework.

Secondly, when training the noise translation net, the so-called "explicit loss" should play an important role since it encode the stochastic knowledge of noises into the learning process. But the aimed labels are a preset Gaussian, with pregiven mean and variance (actually evaluated from the current predicted noise). This is a too ideal pre-assumption in my opinion. If the real noise is very complicated (like non-iid or non-Gaussian), and the pretrained network is not very accurate, I think such a label guidance can not guaranteed to be stably and robustly performed. Not with a solid theoretical support, I think such ideal assumption inclines to make the proposed method too heuristic and applicable only in a narrow domain.

Thirdly, when handling complicated real noise cases, the proposed method seems to be highly dependent on the pre-collected noisy and clean image pairs. This, however, should be extremely difficult in more real scenarios, that is, in real applications, it is generally difficult to achieve large amount high-quality ground truth clean images for those real noisy ones. In this sense, I think the manner used in Dual Adversarial Network or other unsupervised manners should be more useful and meaningful.

**Questions:**

Please see my "weakness" part.

---

> ### Author Response · Authors · 2024-11-13
>
> Dear reviewer 6pYh,
>
> Thank you for your detailed review and feedback on our paper.
> We have posted the rebuttal addressing each of your comments and concerns.
> We appreciate your constructive insights, which have guided us to enhance the clarity and depth of our paper.
>
> Thank you once again for your time and valuable feedback.
>
> Best regards,
> The Authors

---

### Author Response · Authors · 2024-11-13
**Global Rebuttal 1**

### [Reviewer 6pYh, UMZZ, jgKR] Additional comparison with related works

[LAN: Learning to Adapt Noise for Image Denoising]

We would like to emphasize that LAN is a self-supervised method, which adds learned parameters to the input noisy image by backpropagating 5~20 iterations per every single image. In contrast, our method translates noise to the target distribution with minimal computational complexity using a lightweight noise translation network. Due to the computational burden, the LAN paper mostly performed evaluations at a 256x256 resolution. In contrast, our approach is much less computationally intensive, as it requires no additional iterations and uses a lightweight noise translation network, allowing for 1024x1024 inference in a single pass on Restormer.
For quantitative comparison with LAN, we evaluate our method under the same setting with LAN at a 256x256 resolution in the PolyU dataset. Our framework achieves 39.34dB with Restormer, which obviously exceeds the maximum performance of LAN (39.30dB with Restormer, 10 iterations).
We will add more detailed discussion and analysis with LAN in the revised version.


[Dual Adversarial Network: Toward Real Noise Removal and Noise Generation]

As Reviewer 6pYh suggested, our noise translation and denoising network could be trained simultaneously. However, we highlight that a key contribution of our work is the efficient adaptation of a pre-trained denoising model through a lightweight noise translation network. Note that training our noise translation network, in contrast to training an entire network from scratch, requires only 0.5-1% of the computational cost.

Additionally, we compared the performance of our framework with DANet, as shown in the table below. Our method consistently outperforms DANet on most OOD datasets, while DANet even falls short compared to Restormer baseline (OOD Avg PSNR of 38.85 dB)


|           |  SIDD   |  Poly   |   CC    | HighISO | iPhone | Huawei |  OPPO  |  Sony  | Xiaomi | OOD Avg |
|-----------|:---------:|:---------:|:---------:|:---------:|:--------:|:--------:|:--------:|:--------:|:--------:|:---------|
| DANet | **39.46** |  37.53  |  36.15  |  38.34  | 40.45  |  38.40  |  39.85  | **44.35** |  35.57  |  38.83  |
| Ours  |  39.17  | **38.67** | **37.82** | **39.83** | **41.94** | **39.71** | **40.45** |  44.17  | **36.14** | **39.86** |

---

### Author Response · Authors · 2024-11-13
**Global Rebuttal 2**

### [Reviewer Nvpd, UMZZ] Is it possible for a single noise translation network to pair with a different denoising network?

To validate the generalization ability of our noise translation network, we first train a noise translation network in combination with a pre-trained NAFNet. We then evaluate its performance by pairing the noise translation network with a pre-trained Restormer. As shown in the below table, this cross-performance still remains high, demonstrating the network's generality.

|           | Sidd  | Poly  |  CC   | HighISO | iPhone | Huawei | OPPO  | Sony  | Xiaomi | OOD Avg. |
|-----------|:-----:|:-----:|:-----:|:-------:|:------:|:------:|:-----:|:-----:|:------:|:-------:|
| NAF-Res   | 39.19 | 38.64 | 37.27 |  39.95  |  41.93 |  39.64 | 40.51 | 43.99 |  36.11 |  39.76  |
| NAF-NAF   | 39.17 | 38.67 | 37.82 |  39.94  |  41.94 |  39.74 | 40.45 | 44.17 |  36.14 |  39.86  |

---

### Author Response · Authors · 2024-11-13
**Global Rebuttal 3**

### [Reviewer jgKR] Is explicit loss necessary for the performance?

Although our method seems to work well with baseline translation only with Gaussian injection terms, the optimal performance is achieved when all terms are applied. We emphasize that achieving higher PSNR is a lot more difficult when the performance is already quite high. The average performance gain on the OOD datasets from Gaussian injection is 0.34 dB, and the gain from the explicit loss is 0.25 dB, indicating that the contribution from explicit loss is also significant.

### [Reviewer UMZZ] Is L_freq component in explicit loss necessary?

We introduce L_freq because element-wise Gaussian noise does not necessarily guarantee spatial uncorrelation. To illustrate this, consider 2D Gaussian noise after applying a 3x3 averaging kernel. The result retains spatial correlation, even though it follows a Gaussian distribution element-wise in the spatial domain.

---

### Author Response · Authors · 2024-11-13
**Global Rebuttal 4**

### [Reviewer jgKR, k3tR] Is this evaluation setting realistic? Why not try translating from synthetic noise?

Our research addresses real-world noise, which is challenging to model with a specific distribution due to various factors such as camera settings and scene conditions. In our experiments, each real-world noise dataset exhibits a unique distribution, so training on a specific dataset does not consistently enhance performance on others. In this context, good generalization is the ability to adapt effectively to any realistic noise distribution. Therefore, we do not need to assume that real-world noise follows a specific distribution, such as Poisson noise, as our setting is already obviously realistic. We empirically validate that our framework can handle arbitrary noise types within a realistic scope, including Poisson-distributed noise.


### [Reviewer Nvpd, k3tR] Is training with only the SIDD dataset sufficient?

We first refer to Figure 9 in our supplementary document to verify its effectiveness on unseen real images. As presented, our framework trained solely on SIDD dataset performs well even on the images captured by smartphone. This success is due to (1) noise translation being more generalizable than direct denoising (less prone to overfitting) and (2) the use of noise injection during both training and inference, which enhances our framework’s robustness against noise type shifts at test time.

---

### Note · Authors · 2024-11-15

I have read and agree with the venue's withdrawal policy on behalf of myself and my co-authors.